

# Allelopathic effects and composition of aqueous extracts from different parts of *Galinsoga parviflora* Cav. on *Medicago sativa* L. and *Avena sativa* L

Shipu Cheng[1,*], Fanru Xu[1,*], Zhiyong Lu[1], Huairui Xu[1], Mengqi Cai[1], Juan Sun[1,2] and Yufang Xu[1,2]

[1] College of Grassland Science, Qingdao Agricultural University, Qingdao, Shandong, China
[2] Shandong Key Laboratory for Germplasm Innovation of Saline-alkaline Tolerant Grasses and Trees, Qingdao, Shandong, China
[*] These authors contributed equally to this work.

Corresponding author
Yufang Xu, xuyufang@qau.edu.cn

## ABSTRACT

**Background**. *Galinsoga parviflora* Cav. is a high-risk invasive plant that seriously threatens the development of grasslands in southern China. However, the allelopathic effects on *Medicago sativa* L. and *Avena sativa* L., which are widely cultivated forages around the world, have not been reported.

**Methods**. To explore the allelopathic mechanism of *G. parviflora*, the allelopathic effects of aqueous extracts from different parts of *G. parviflora* on *M. sativa* and *A. sativa* were investigated. The germination rate (GR), germination potential (GP), seedling height, fresh weight, and chlorophyll content of *M. sativa* and *A. sativa* seedlings were measured to elucidate the allelopathy of *G. parviflora* on the two forages. Based on the five indicators, synthetical allelopathic effects (SAE) of extracts was also calculated. In addition, the allelopathic components of the extracts in *G. parviflora* were quantitatively revealed by untargeted metabolomics detection. Furthermore, two key allelopathic substances, 1,4-cyclohexanedicarboxylic acid (CHDA) and trehalose, were selected to explore the inhibitory effect on two notorious weed species in China, such as gramineous *Digitaria sanguinalis* L. and broad-leaved *Amaranthus retroflexus* L.

**Result**. (1) The inhibitory effects of aqueous extracts from different parts of *G. parviflora* on recipient plants were different, the root was the weakest, and the whole plants was the strongest, with the values of synthetical allelopathic effects (SAE) on *M. sativa* at the highest concentration being −0.12 and −0.40, respectively. (2) Compared with *A. sativa*, *M. sativa* was generally more susceptible to the extracts. (3) The differences in the content of CHDA or trehalose might be a reason why extracts from different parts of *G. parviflora* exhibited different allelopathic effects. (4) The herbicidal activity test of key allelopathic substances found that CHDA has a strong inhibitory effect on the germination of *D. sanguinalis* and almost does not affect *M. sativa* and *A. sativa*. Thus, this discovery not only revealed allelopathic effects and components in different parts of *G. parviflora*, but provided scientific evidence for weed control based on natural plant extracts in the future.

## INTRODUCTION

*Galinsoga parviflora* Cav., also known as smallflower galinsoga or gallant soldier, belongs to the Asteraceae genus *Galinsoga*. This plant was native to Central and South America, but now it was reported in over 32 crops in 38 countries (*Paula et al., 2022*). As a high-risk invasive plant, *G. parviflora* with strong reproductive and competitive abilities had become one of the most notorious weeds in meadows, roadside, ditches, wastelands, orchards and vegetable fields in worldwide (*Mozdzen et al., 2018*; *Xie et al., 2021*). In addition, *G. parviflora* was also considered to be a host of plant pathogens, such as *Pseudomonas marginalis* pv. marginalis (*Paula et al., 2022*; *Damalas, 2008*). At present, *G. parviflora* has been included in the list of alien invasive species of China and belongs to the category of severe invasion, which means that it has caused significant losses and impacts on economic and ecological benefits at the provincial level (*Gao, Sun & Jiang, 2021*).

Allelopathy, also called the interaction effect, referred to an interference mechanism in which the plant release bioactive metabolites that had direct or indirect beneficial or harmful effects on surrounding plants and microorganisms (*Qu et al., 2021*). Allelochemicals released by root exudates, transpiratory secretions and the results of the decomposition of their residues in the soil (*Tsytsiura & Sampietro, 2024*; *Scavo, Abbate & Mauromicale, 2019*). Meanwhile, many studies also suggested that allelopathy might play an essential role in the successful invasion of exotic plants. Some invasive plants can disrupt the germination and seedling growth of native plants by secreting allelochemicals to gain a competitive advantage (*Bais et al., 2003*; *Cai et al., 2024*). Therefore, the exploration of allelopathic effects was also crucial for controlling invasive *G. parviflora*.

In fact, studies of the allelopathic effects of *G. parviflora* on some plants were gradually reported. For example, extracts from *G. parviflora* at higher concentration showed significantly inhibitory effects on the seed germination and seedling growth of *Raphanus sativus* L. (*Mozdzen et al., 2018*; *Tsytsiura & Sampietro, 2024*). *G. parviflora* extracts inhibited the seedling growth of *Vicia faba* L. by inhibiting the ability of assimilation of $CO_2$ and reducing the biomass (*Huang et al., 2017*). In addition, *G. parviflora* also exhibited definite allelopathic potential on plants such as *Geum japonicum* Thunb. var chinense, *Trifolium repens* L. and *Fagopyrum esculentum* Moench. (*Xie et al., 2021*; *Wang et al., 2012*; *Yao et al., 2014*). However, there have been no report about the allelopathy of *G. parviflora* on *Medicago sativa* L. and *Avena sativa* L. They are not only widely cultivated forages in the world, but their intercropping and mixed sowing patterns are also important cultivation modes (*Liu et al., 2023*; *Spaner & Todd, 2004*). At present, *G. parviflora* has become one of the notorious weeds in forage fields, seriously threatening the feed quality in worldwide (*Zhao, 2021*; *Riemens & Weide, 2008*). Thus, exploring the allelopathic effects of invasive *G. parviflora* on *M. sativa* and *A. sativa* may lay a good foundation for research on the stability and sustainable utilization of artificially cultivated grasslands.

Not only that, invasive plants with allelopathy were often found to have herbicidal potential, such as *Artemisia argyi* L. and *Acacia dealbata* L., providing a new perspective for weed control in the future (*Li et al., 2021*; *Souza-Alonso et al., 2020*). Ulteriorly, allelochemicals with herbicidal activity were also considered as the perfect substitute

for traditional herbicides. *Digitaria sanguinalis* L. and *Amaranthus retroflexus* L. were malignant gramineous and broad-leaved weeds worldwide, respectively (*Guan et al., 2024*; *Mitich, 1997*), and which seriously restricted the yield and quality of *M. sativa* in China (*Yang et al., 2022*). As a result, the herbicidal activity of key allelochemicals on the two weeds might be enlightening for the development of plant-derived herbicides.

The purpose of the study was to (1) explore the allelopathy of extracts from different parts of *G. parviflora* on two common forages, (2) illustrate the main allelopathic components, and (3) evaluate herbicidal activity of key allelopathic substances on *D. sanguinalis* and *A. retroflexus*. In summary, this study not only revealed allelopathic effects and components in *G. parviflora*, but these phytotoxic chemicals might be potential sources of bioherbicides in artificially cultivated grasslands and agrocenoses.

## MATERIALS & METHODS

### Preparation of aqueous extracts from roots, stems, leaves and whole plants of it *G. parviflora*

In 2022, seeds of *G. parviflora* were collected from *M. sativa* fields in Qujing City of China (103.21°E, 25.18°N), which were germinated in petri dishes at room temperature for 72 h after rinsing with water. Twelve germinated seeds were planted in a pot with a diameter of 30 cm containing moist loam soil, and kept in artificial climate chambers at 25/20 °C (light/dark) for a 16-h photoperiod with a photosynthetic photon flux density of approximately 270 $\mu$mol m$^{-2}$ s$^{-1}$. During the cultivation process, the soil was watered moderately every five days to ensure moisture.

Approximately 45 d after planting, nearly all *G. parviflora* were in flowering phase, which for most species was the period of maximum allelopathic activity (*Singh, Singh & Singh, 2015*). Meanwhile, the fresh roots, stems, leaves and whole plants of *G. parviflora* were washed, dried and stored at 4 °C after crushing to a particle size of less than 0.5 mm, respectively. Accurate 50 g of the above sample from different parts was added into 500 mL distilled water and then was oscillated for 48 h at the speed of 150 rpm min$^{-1}$ at 4 °C. After 15 min of ultrasound, the mixture was repeatedly filtrated by vacuum with 0.45-$\mu$m membrane to obtain an extraction solution with a final concentration of 0.100 g mL$^{-1}$. Subsequently, the extracts from different parts of *G. parviflora* were diluted with distilled water to solutions with the concentration of 0.025, 0.050, 0.075 and 0.100 g mL$^{-1}$ (*Dai et al., 2022*). The range of concentration was based on a preliminary study, which showed that extracts exhibited a promoting or slight inhibitory effect on recipient plants at the lowest concentration (0.025 g mL$^{-1}$).

### Seed germination test

The varieties of *M. sativa* and *A. sativa* tested were Zhongmu No.3 and Qinghai No.444, respectively, obtained by Beijing Best Grass Industry Co., Ltd in 2023. The seeds were washed for three times, then sterilized with 75% alcohol for one minute, and cleaned with sterile water for five times to remove the alcohol. A total of 50 disinfected seeds of *M. sativa* and *A. sativa* were placed in a square plastic culture dish with a size of 12 × 12 cm (Nantong Mingcheng Experimental Instrument Co., Ltd). Subsequently, eight mL of

different concentrations of extracts from the roots, stems, leaves and whole plants of *G. parviflora* were added. The sealed culture dish was placed under the condition of 25 °C temperature, 15000 lx light intensity, and 12 h/d light duration for cultivation.

During the cultivation, the number of germinated seeds was counted every day, and four mL of corresponding solutions were periodically supplemented to each culture dish every other day. The control group was treated with equal volume of sterile water. The experiments were conducted with four replicates per dose. The seed that produced at least one mm radicle was considered as germinated (*ISTA, 2020*; *Pamplona et al., 2020*). The last count was conducted when no new germinated for three consecutive days. The germination rate (GR) and germination potential (GP) of *M. sativa* or *A. sativa* were calculated according to the following formulas (*Sun et al., 2021*).

GR = Number of germinated seeds within 7 days/total number of test seeds
GP = Number of germinated seeds within 3 days/total number of test seeds.

### Seedling growth test

Ten newly sprouted seeds were put in a culture dish containing above eight mL of extracts, and the method of cultivation was same as mentioned above. Similarly, four mL of the corresponding solution was added to each culture dish every other day, and 48 mL of solution was needed for each treatment during the cultivation. After 21 days of cultivation, ten seedlings were randomly selected from each dish, gently wiped the water on the surface, and their height and fresh weight were measured (*García-Locascio, Valenzuela & Cervantes-Avilés, 2024*). The experiments were conducted with four replicates per dose.

### Determination of chlorophyll content

The leaves in the section of seedling growth test were removed and washed with distilled water to determine the content of chlorophyll. Accurately 200 mg leave tissues in different treatment with 10 mL of 95% ethanol was placed in a dark room for 48 h, and was shaken for three times during the extraction. The absorbance of extracted solution was measured at 665 nm and 649 nm with 95% ethanol as the blank. The content of chlorophyll was calculated based on the following formula (*Dai et al., 2022*).

Chlorophyll a (mg L$^{-1}$) = 13.95 A665 nm − 6.88 A649 nm
Chlorophyll b (mg L$^{-1}$) = 24.96 A649 nm − 7.32 A665 nm
Chlorophyll content (mg g$^{-1}$) = (Concentration of chlorophyll × total volume dilution of the extract)/fresh weight of the samples.

### Analysis of synthetical allelopathic effects

The response index (RI) of extracts to recipient plants was calculated according to the following formula. In this equal, C and T represented the values of control group and treatment group respectively. If the RI value was positive, the effect of extracts was promoted. On the contrary, it was inhibited. The absolute value of RI reflected the magnitude of the allelopathy. The synthetical allelopathic effects (SAE) represented the arithmetic mean of the RI of five measurement index of *M. sativa* or *A. sativa* (*Dai et al., 2022*), including GR, GP, the height and fresh weight of seedings and the content of chlorophyll.

$$RI = 1 - C/T \ (T \geq C)$$
$$RI = T/C - 1 \ (T < C).$$

## Identification of allelopathic substances of *G. parviflora*

In order to further elucidate the mechanism of allelopathic effect, the components of the extracts from different parts of *G. parviflora* were identified. Accurately 100 mg of freshly thawed powders was extracted with one mL pre-cooling extraction solution (water-acetonitrile-isopropyl alcohol mixed solution 1:1:1, v/v/v). After 30 min ultrasound at 4 °C, the mixture was centrifuged at 12,000 rpm for 10 min (X3R, Thermo Fisher Scientific, USA), set still for 1 h to precipitate protein at −20 °C, and then centrifuged 10 min with 12,000 rpm at 4 °C. Subsequently, supernatant was transferred to a centrifuge tube and dried in vacuum. The dried supernatant was redissolved with 200 µL of 30% acetonitrile (Sinopharm Chemical Reagent Co., Ltd, EC number: 200-835-2) solution, centrifuged 15 min with 14,000 rpm at 4 ° C. Finally, the supernatant was filtered through and analyzed by ultra high performance liquid chromatography (UHPLC, Vanquish; Thermo Fisher Scientific, Waltham, MA, USA) and high resolution mass spectrum (HRMS, Q Exactive HFX; Thermo Fisher Scientific) (*Ghirardo et al., 2020*). The composition analysis was conducted by UHPLC with a Waters HSS T3 column (100 × 2.1 mm, 1.8 µm) with a temperature of 40 °C. Mobile phase A was Mili-Q water (0.1% formic acid) (Sinopharm Chemical Reagent Co., Ltd, EC number: 200-579-1), and phase B was acetonitrile (0.1% formic acid). The original proportion of phase A/phase B was 100:0 (v/v) in the first 12 min, changed to 5:95 (v/v) within 12-13 min, and returned to original 100:0 (v/v) from 13.1 min to the end (17 min). Data of HRMS were recorded by the Full-ms-ddMS2 acquisition methods. The ESI source parameters were set as follows: sheath gas pressure, 40 arb; aux gas pressure, 10 arb; spray voltage, +3000 v/-2800 v; temperature, 350 °C; and ion transport tube temperature, 320 °C. The scanning range of primary MS was (scan m/z range) 70–1050 Da, with a primary resolution of 70,000 and secondary resolution of 17,500.

## Determination of herbicidal activity of key allelopathic substances

According to the identification of extracts from *G. parviflora*, the content of 1,4-cyclohexanedicarboxylic acid (CHDA) and trehalose ranked the top two. At the same time, they were also the highest in all phenolic acids or polysaccharides, and were selected to explore the herbicidal activity of key allelopathic substances. Technical-grade CHDA (EC number: 214-068-6) and trehalose (EC number: 202-739-6) were purchased in Shanghai Yuanye Bio-Technology Co., Ltd. The stock solution with a concentration of 1.0 g mL$^{-1}$ was prepared by dissolving the above substance in distilled water and stored at 4 °C for later use. Subsequently, distilled water was added to dilute the CHDA solution to seven concentrations, namely 1.0, 0.1, $1.0 \times 10^{-2}$, $5.0 \times 10^{-3}$, $1.0 \times 10^{-3}$, $1.0 \times 10^{-4}$ and $1.0 \times 10^{-5}$ g mL$^{-1}$. Due to the lower phytotoxicity of trehalose, only five solutions with higher concentrations mentioned above were prepared.

The seeds of *D. sanguinalis* and *A. retroflexus* to determine herbicidal activity were collected from *M. sativa* fields in Qingdao City, China (120.08°E, 36.45°N) in September

2022. To break dormancy of the two weeds, the harvested seeds were soaked in 98% concentrated sulfuric acid (Sinopharm Chemical Reagent Co., Ltd, EC number: 231-639-5) for two minutes, and cleaned with sterile water for three times before use. Meanwhile, the above-mentioned *M. sativa* and *A. sativa* were also selected to evaluate safety of the two allelopathic substances. The method for seed germination testing was consistent with that mentioned earlier.

## Statistical analysis

Data was submitted to Shapiro–Wilk's normality test. When the assumptions of normality were met, the *Dunnett*'s test ($P < 0.05$) at the 5% level of significance in one-way analysis of variance (ANOVA) was carried out to compare the RI values under five concentrations (C0-C4) and four parts (P1-P4) in the same indicator. Meanwhile, the two-way ANOVA was also conducted on the RI values of each indicator to analyze effects of part (P), concentration (C), and their interactions (P × C). In terms of herbicidal activity, the differences between different concentrations of CHDA or trehalose solutions were compared by one-way ANOVA. Statistical analysis was performed using SPSS software (*v.* 20.0, IBM, USA), and the data was expressed as mean ± standard deviation (SD).

## RESULTS

### Effects of extracts from different parts of *G. parviflora* on the germination of *M. sativa* and *A. sativa*

Both GR and GP were used to evaluate the allelopathic effects of the extracts on the germination of *M. sativa* and *A. sativa* (Fig. 1). The extracts of *G. parviflora* displayed varied degrees of inhibitory effects, and the concentrations of extracts and parts of *G. parviflora* exhibited significant effects on the GR of *M. sativa* and *A. sativa* ($P < 0.001$, Figs. 1A, 1B). The allelopathic effects of extracts gradually strengthened with the increase of concentration. When the concentration was 0.075 g mL$^{-1}$, the GR of *M. sativa* significantly decreased under the extracts from various parts of *G. parviflora* (Fig. 1A). The inhibitory effect was the strongest at the concentration of 0.100 g mL$^{-1}$, and the GR of *M. sativa* was only 48% treated by extracts from the whole plants (Fig. 1A). Compared with *M. sativa*, *A. sativa* showed a higher tolerance to extracts of *G. parviflora*, the higher concentration (0.100 g mL$^{-1}$) of corresponding extracts were required to significantly inhibit its germination (Fig. 1B). In addition, the allelopathic effects of extracts from different parts of *G. parviflora* with the same concentration were different. There was no significant difference in the effect of extracts from different parts of *G. parviflora* on the GR of *M. sativa* and *A. sativa* at the lowest concentration (0.025 g mL$^{-1}$), and the differences gradually emerged accompanied by the increase in concentration. Overall, the allelopathic ability from weak to strong was roots < stems < leaves < whole plants, and the corresponding RI values at the highest concentration were −0.07, −0.14, −0.39, and −0.49, respectively (Fig. 1A). Nevertheless, there was no significant difference in allelopathic effects of extracts from the leaves and other two parts on *A. sativa* (Fig. 1B).

The concentrations of extracts, parts of *G. parviflora* and their interactions also exhibited significant effects on GP of *M. sativa* ($P < 0.001$, Fig. 1C) and *A. sativa* ($P < 0.01$, Fig. 1D),

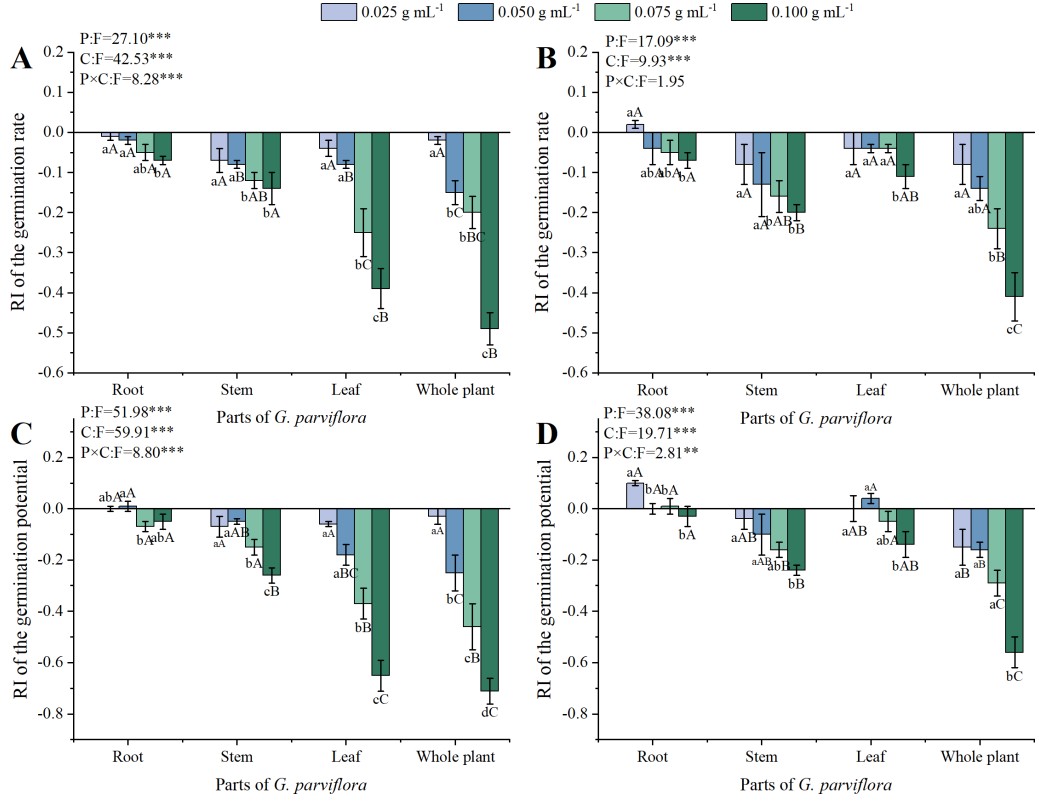

**Figure 1** Effects of different extracts from the root, stem, leaf and whole plant of *Galinsoga parviflora* Cav. on the germination rate (A, B) and germination potential (C, D) of *Medicago sativa* L. and *Avena sativa* L. Data is shown as the mean ± standard deviation (SD) of four replicates. Different lowercase letters indicate significant differences between extracts of four concentrations (C) from the same part of *G. parviflora* ($P < 0.05$). Different capital letters indicate significant differences between extracts of four parts (P) at the same concentration ($P < 0.05$). The symbol of "*" represents significant differences between the treatments of parts (P), concentrations (C), and their interactions (P × C). One asterisk (*) indicate $P < 0:05$, two asterisks (**) indicate $P < 0:01$, three asterisks (***) indicate $P < 0:001$.

which was highly correlated with GR, with a Pearson correlation coefficient of 0.962. Similarly, the allelopathic effect of extracts on GP gradually increased with increasing concentration, and all different parts of the extracts showed significant difference on the GP of *M. sativa* at the concentration of 0.100 g mL$^{-1}$ (Fig. 1C). The influence of different parts of extracts from *G. parviflora* on the GP of recipient plants was also different. The inhibitory effect of extracts from the whole plants of *G. parviflora* was the strongest, the RI values of *M. sativa* and *A. sativa* at the highest concentration were −0.71 and −0.56, respectively (Figs. 1C, 1D). On the contrary, the inhibitory effect of root extracts was the weakest, and no significant inhibitory effect on the GP of recipient plant seeds at various concentrations was observed (Fig. 1).

### Effects of extracts from different parts of *G. parviflora* on the seedling growth of *M. sativa* and *A. sativa*

The concentrations and parts also had significant effects on the seedling growth of *M. sativa* and *A. sativa* ($P < 0.001$, Fig. 2). The extracts of *G. parviflora* exhibited promoting growth at low concentrations and inhibiting growth at high concentrations on the seedling height of *M. sativa* (Fig. 2A), which was different from the allelopathic effect on the germination of recipient plants (Fig. 1). Almost all extracts with the concentration of 0.025 g mL$^{-1}$ promoted the height of *M. sativa* seedlings. It was not until the concentration reached 0.075 g mL$^{-1}$ that the extracts from various parts began to show significant inhibitory effects on the height of *M. sativa*. The inhibitory was most significant at the concentration of 0.100 g mL$^{-1}$ (Fig. 2A). However, the promotion effect of extracts from various parts on the growth of *A. sativa* was not significant at low concentration, and the inhibitory effect only became apparent when the concentration of the extracts was 0.075 g mL$^{-1}$ (Fig. 2B). Similarly, the allelopathic effect of root extracts on the height of *M. sativa* and *A. sativa* plants was significantly lower than that from other parts of *G. parviflora*, and the values of RI at the concentration of 0.100 g mL$^{-1}$ was $-0.08$ and $-0.40$, respectively. In addition, the sensitivity of *M. sativa* and *A. sativa* to extracts from different parts of *G. parviflora* was also varied. For instance, the inhibitory effect of stem extracts on *A. sativa* height was far stronger than that of leaf extracts at the concentration of 0.100 g mL$^{-1}$, with corresponding RI values of $-0.64$ and $-0.49$, while no significant difference was observed on the height of *M. sativa* (Figs. 2A, 2B).

In addition to height, fresh weight was also used in this study to evaluate the allelopathic effect of the extracts on the seedling growth of recipient plants (Figs. 2C, 2D). Compared with height, the inhibitory effect of the extracts of *G. parviflora* on the fresh weight of seedlings was weaker. For example, all concentrations of stem extracts showed no significant inhibitory effect on the fresh weight of *M. sativa*. The extracts from all parts of *G. parviflora* at the concentration of 0.025 g mL$^{-1}$ or 0.050 g mL$^{-1}$ exhibited a positive effect on the fresh weight of *M. sativa*, which was similar to the results of plant height (Fig. 2C). Only extracts from the whole plants of *G. parviflora* exhibited a significant inhibitory effect on the fresh weight of *A. sativa* at the highest concentration, while the extracts from other parts did not (Fig. 2D). This phenomenon once again indicated the differences in allelopathic effects of different parts from *G. parviflora*.

### Effects of extracts from different parts of *G. parviflora* on the chlorophyll content of *M. sativa* and *A. sativa*

The extracts of *G. parviflora* exhibited varied degrees of allelopathic effects on the chlorophyll content of *M. sativa*, and the allelopathic effects was also affected by the concentrations (C) of extracts, and parts (P) of *G. parviflora* ($P < 0.001$, Fig. 3A). The RI values of roots, stems, leaves, and whole plants extracts at the concentration of 0.100 g mL$^{-1}$ on the chlorophyll content were $-0.18$, $-0.47$, $-0.41$ and $-0.51$ (Fig. 3A), respectively, which was basically consistent with the results of germination and seedling growth mentioned earlier (Figs. 1–2). However, the inhibitory effect of extracts on the chlorophyll content of *A. sativa* was relatively weaker, and there was no significant difference

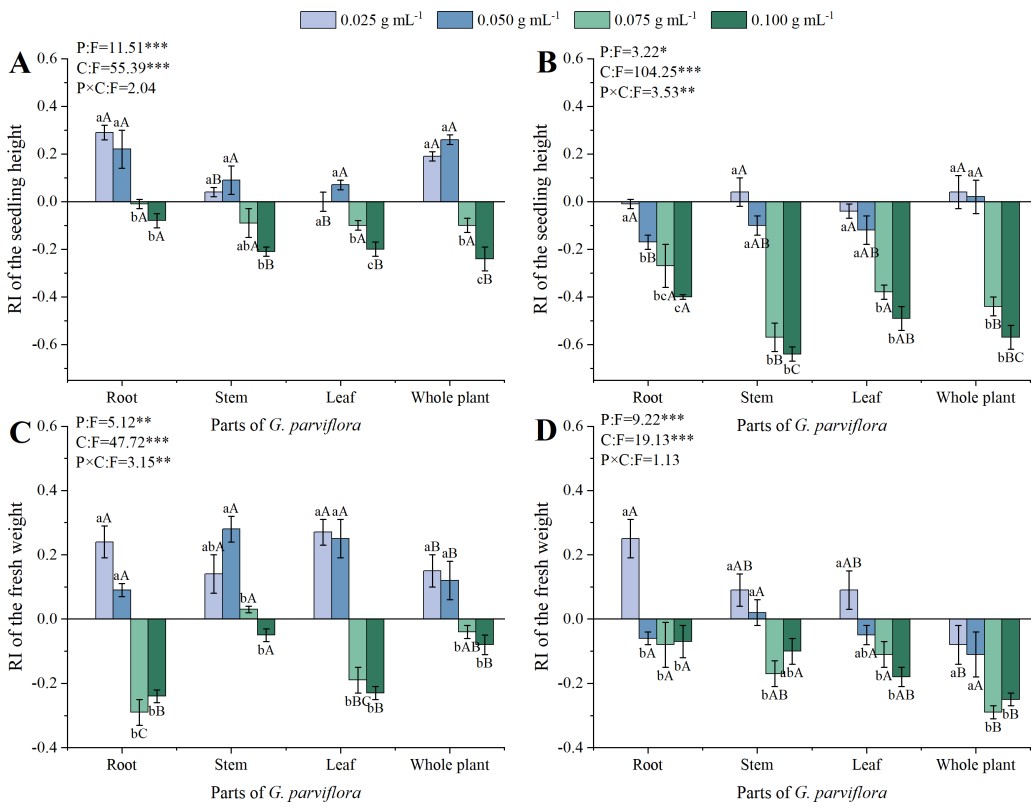

**Figure 2** **Effects of different extracts from the root, stem, leaf and whole plant of *G. parviflora* on the height (A, B) and fresh weight (C, D) of *M. sativa* and *A. sativa* seedlings.** One asterisk (*) indicate $P < 0.05$, two asterisks (**) indicate $P < 0.01$, three asterisks (***) indicate $P < 0.001$.

between the extracts of different concentrations (Fig. 3B). This might be because inhibiting the chlorophyll synthesis in *A. sativa* required a concentration of *G. parviflora* extracts exceeding 0.100 g mL$^{-1}$.

## Synthetical allelopathic effects of extracts from different parts of *G. parviflora* on *M. sativa* and *A. sativa*

The SAE values of extracts from different parts of *G. parviflora* were calculated based on the above five indicators and shown in Fig. 4. The concentrations of extracts, parts of *G. parviflora* and their interactions significantly affected the SAE of *M. sativa* and *A. sativa* ($P < 0.001$, Fig. 4). All extracts with concentrations exceeding 0.050 g mL$^{-1}$ exhibited inhibitory effects on both recipient plants (SAE<0), and the allelopathic effect gradually increased with increasing concentrations. The allelopathic effects of the extracts from different parts of *G. parviflora* varied greatly. The effect of roots was the weakest, while the whole plants was the strongest, and the SAE values at the concentration of 0.100 g mL$^{-1}$ to *M. sativa* was −0.12 and −0.40, respectively (Fig. 4A). The allelopathic effects of the extracts from stems and leaves were moderate, and their intensity on the two types of forage was opposite. According to the values of SAE at the low concentration, *M. sativa* was generally more sensitive to extracts than *A. sativa* (Fig. 4).

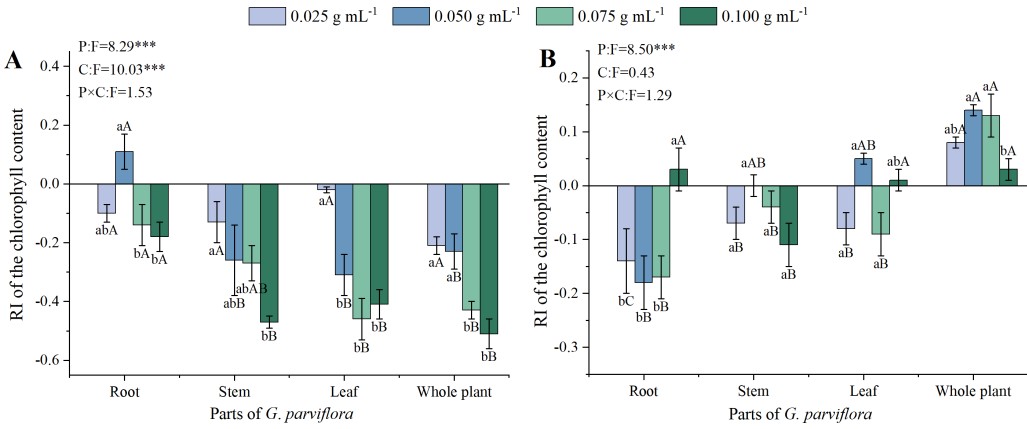

**Figure 3** Effects of different extracts from the root, stem, leaf and whole plant of *G. parviflora* on the chlorophyll content of *M. sativa* (A) and *A. sativa* (B). One asterisk (*) indicate $P < 0:05$, two asterisks (**) indicate $P < 0:01$, three asterisks (***) indicate $P < 0:001$.

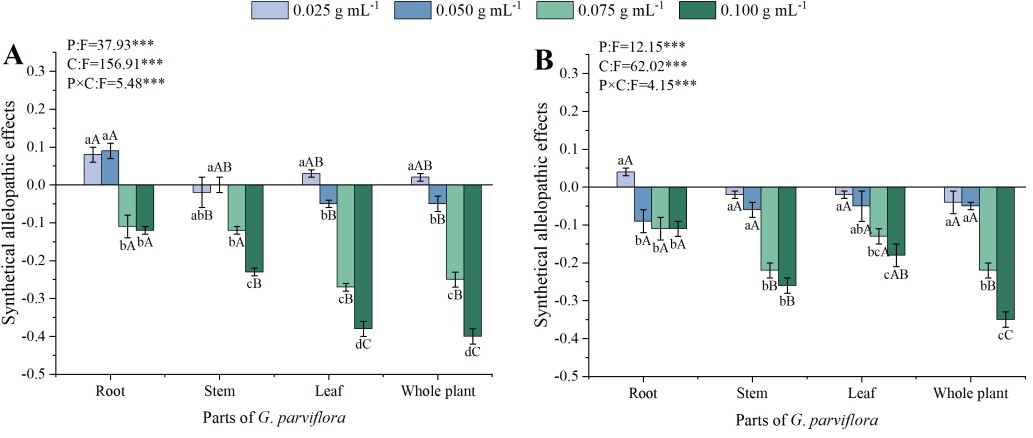

**Figure 4** The synthetical allelopathic effects (SAE) of extracts from the root, stem, leaf and whole plant of *G. parviflora* on *M. sativa* (A) and *A. sativa* (B). One asterisk (*) indicate $P < 0:05$, two asterisks (**) indicate $P < 0:01$, three asterisks (***) indicate $P < 0:001$.

## Identification of allelopathic substances of *G. parviflora*

To analyze the mechanism of the allelopathic effects of extracts from different parts of *G. parviflora* on recipient plants, untargeted metabolomics detection was conducted. The number of chemical components detected in the extracts of the roots, stems, leaves, and whole plants of *G. parviflora* is 305, 310, 308, and 313, respectively, including acids, sugars, glycosides, terpenoids, etc. The composition and content in the extracts from different parts varied wildly (Fig. 5), and detailed information, such as the chemical name, type, formula, CAS number, fragmentation score, retention time ($t_R$) of the extracts from the whole plants were listed in Table 1. Obviously, the component with the highest content in the extracts from the whole plants, stems and leaves was CHDA, with relative contents of 29.65%, 34.69% and 37.39%, respectively. However, this component was rarely found
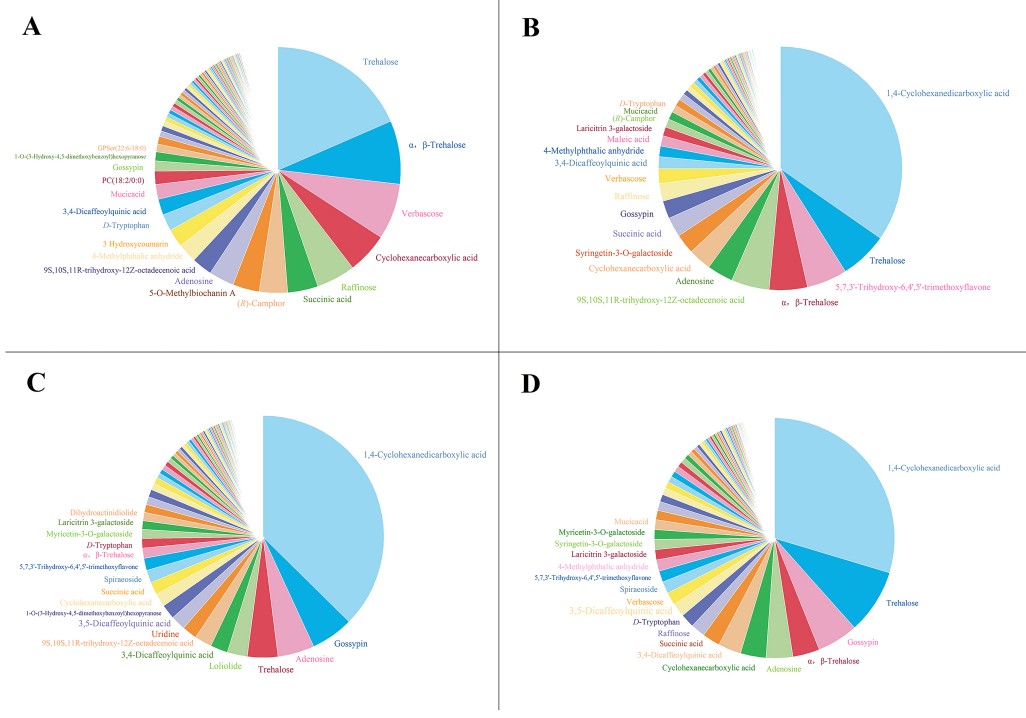

**Figure 5  The composition of the extracts from the root (A), stem (B), leaf (C) and whole plant (D) of _G. parviflora._** Substances with the content exceeding 1% were listed.

in root extracts, and the highest content was occupied by 18.49% trehalose. The trehalose content detected in extracts from the whole plants and stems was only second to CHDA, with the values of 8.76% and 6.36%, respectively. The content of trehalose in leaf extracts was also 4.89%, which indicated that _G. parviflora_ was a plant rich in trehalose. In addition, components such as gossypin, succinic acid and 9$S$,10$S$,11$R$-trihydroxy-12$Z$-octadecenoic acid were commonly present in the extracts from various parts of _G. parviflora_ (Fig. 5).

## Determination of herbicidal activity of key allelopathic substances

The effects of CHDA and trehalose on the GR of _D. sanguinalis_, _A. retroflexus_, _M. sativa_ and _A. sativa_ were shown in Table 2. The CHDA significantly inhibited the germination of _D. sanguinalis_ starting from the concentration of $1.0 \times 10^{-4}$ g mL$^{-1}$ with a GR of 60%. And CHDA showed a hormesis effect on the germination of _A. retroflexus_, with a GR increased by 24% at the concentration of $1.0 \times 10^{-3}$ g mL$^{-1}$ and almost no germination at the concentration of $5.0 \times 10^{-3}$ g mL$^{-1}$. In terms of _M. sativa_ and _A. sativa_, CHDA had no significant effect on their GRs at low concentrations, while reduced GRs to 49% and 59% at the highest concentration (1.0 g mL$^{-1}$), respectively. Meanwhile, trehalose does not affect the germination of the four tested plants (Table 2).

**Table 1 Composition of the extracts from the whole plants of *Galinsoga parviflora*.**

| No | Name | Type | Formula | CAS | Fragmentation score | $t_R$ (min) |
|---|---|---|---|---|---|---|
| 1 | 1,4-Cyclohexanedicarboxylic acid | Dicarboxylic acid | $C_8H_{12}O_4$ | 1076-97-7 | 80.9 | 6.88 |
| 2 | Trehalose | Disaccharide | $C_{12}H_{22}O_{11}$ | 99-20-7 | 96.6 | 0.79 |
| 3 | Gossypin | Flavonoid | $C_{21}H_{20}O_{13}$ | 540-25-4 | 97.3 | 5.34 |
| 4 | $\alpha, \beta$-Trehalose | Disaccharide | $C_{12}H_{22}O_{11}$ | 585-91-1 | 67.2 | 0.79 |
| 5 | Adenosine | Nucleoside | $C_{10}H_{13}N_5O_4$ | 58-61-7 | 90.4 | 1.43 |
| 6 | Cyclohexanecarboxylic acid | Aromatic acid | $C_{16}H_{18}O_9$ | 630-21-3 | 76.9 | 4.66 |
| 7 | 3,4-Dicaffeoylquinic acid | Hydroxycinnamic acid ester | $C_{25}H_{24}O_{12}$ | 14534-61-3 | 79.6 | 6.17 |
| 8 | Succinic acid | Dicarboxylic acid | $C_4H_6O_4$ | 110-15-6 | 60.8 | 1.20 |
| 9 | Raffinose | Trisaccharide | $C_{18}H_{32}O_{16}$ | 512-69-6 | 95.5 | 0.79 |
| 10 | *D*-Tryptophan | Amino acid | $C_{11}H_{12}N_2O_2$ | 1756-70-5 | 85.7 | 3.64 |
| 11 | 3,5-Dicaffeoylquinic acid | Hydroxycinnamic acid derivative | $C_{25}H_{24}O_{12}$ | 2450-53-5 | 77.5 | 6.07 |
| 12 | Verbascose | Tetrasaccharide | $C_{30}H_{52}O_{26}$ | 585-83-9 | 73.9 | 0.83 |
| 13 | Spiraeoside | Flavonoid glycoside | $C_{21}H_{20}O_{12}$ | 20249-74-3 | 98.3 | 5.72 |
| 14 | 5,7,3′-Trihydroxy-6,4′,5′-Trimethoxyflavone | Flavonoid | $C_{18}H_{16}O_8$ | 78417-26-2 | 69.6 | 8.41 |
| 15 | 4-Methylphthalic anhydride | Anhydride | $C_9H_6O_3$ | 1124-11-4 | 63.5 | 4.66 |
| 16 | Laricitrin 3-galactoside | Flavonoid glycoside | $C_{22}H_{22}O_{13}$ | 93219-26-2 | 80.7 | 5.97 |
| 17 | Syringetin-3-O-galactoside | Flavonoid glycoside | $C_{23}H_{24}O_{13}$ | 55025-56-4 | 84.7 | 6.18 |
| 18 | Myricetin-3-O-galactoside | Flavonoid glycoside | $C_{21}H_{20}O_{13}$ | 15648-86-9 | 85 | 5.35 |
| 19 | Mucicacid | Organic acid | $C_6H_{10}O_8$ | 526-99-8 | 66.3 | 7.35 |

**Notes.**

'No' indicates ranking of content in whole plant extracts of *G. parviflora* from high to low. Substances with the content exceeding 1% were listed, and all substances could be found in Table S1.

## DISCUSSION

The allelopathic substances produced by one plant will be released into the environment and affect various growth stages of surrounding plants, including seed germination, seedling growth and chlorophyll synthesis (*Dai et al., 2022*; *Wang et al., 2022a*; *Wang et al., 2022b*). The results in the study showed that aqueous extracts of *G. parviflora* had hormesis effects, which stimulate at low concentrations but inhibit at high concentrations on the height and fresh weight of seedling and SAE. The promoting effect of extracts from *G. parviflora* at low concentration on the two types of forage was also observed (Fig. 2), which was similar to a previous study on the seedlings of *Lactuca sativa* L. and *R. sativus* (*Wang et al., 2022a*; *Wang et al., 2022b*; *Mozdzen et al., 2018*; *Tsytsiura & Sampietro, 2024*). That may be because that the low concentrations of allelopathic substances, such as p-hydroxybenzoic acid and chlorogenic acid, also have hormesis effects (*Pannacci et al., 2022*). Meanwhile, the inhibitory effects of high concentrations extracts were applicable to all indicators of two common forages (Figs. 1–4). There is no apparent pattern in terms of the stimulatory, inhibitory, or hormesis effects of different indicators.

The allelopathic effects of different parts of plants on recipient plants are generally not exactly the same (*Wang et al., 2024*). This study first determined the differences in

Cheng et al. (2025), *PeerJ*, DOI 10.7717/peerj.19378

**Table 2 Germination rate of *Digitaria sanguinalis*, *Amaranthus retroflexus*, *Medicago sativa* and *Avena sativa* treated with 1,4-cyclohexanedicarboxylic acid or trehalose.**

| Concentration (g mL$^{-1}$) | 1,4-cyclohexanedicarboxylic acid | | | | trehalose | | | |
|---|---|---|---|---|---|---|---|---|
| | *D. sanguinalis* | *A. retroflexus* | *M. sativa* | *A. sativa* | *D. sanguinalis* | *A. retroflexus* | *M. sativa* | *A. sativa* |
| 0 | 0.83 ± 0.06b | 0.58 ± 0.07b | 0.93 ± 0.02a | 0.80 ± 0.03a | 0.83 ± 0.06a | 0.58 ± 0.07a | 0.93 ± 0.02a | 0.80 ± 0.03a |
| 1.0 × 10$^{-5}$ | 0.97 ± 0.03a | 0.54 ± 0.06b | 0.92 ± 0.03a | 0.88 ± 0.07a | ND[a] | ND | ND | ND |
| 1.0 × 10$^{-4}$ | 0.60 ± 0.03c | 0.68 ± 0.04ab | 0.89 ± 0.06a | 0.86 ± 0.04a | ND | ND | ND | ND |
| 1.0 × 10$^{-3}$ | 0.01 ± 0.02d | 0.82 ± 0.02a | 0.92 ± 0.02a | 0.88 ± 0.03a | 0.91 ± 0.10a | 0.68 ± 0.09a | 0.89 ± 0.04a | 0.80 ± 0.05a |
| 5.0 × 10$^{-3}$ | 0.00 ± 0.00d | 0.03 ± 0.02c | 0.90 ± 0.03a | 0.87 ± 0.03a | 0.92 ± 0.02a | 0.51 ± 0.13a | 0.88 ± 0.02a | 0.81 ± 0.04a |
| 1.0 × 10$^{-2}$ | 0.00 ± 0.00d | 0.10 ± 0.06c | 0.90 ± 0.05a | 0.83 ± 0.05a | 0.85 ± 0.06a | 0.56 ± 0.15a | 0.91 ± 0.04a | 0.79 ± 0.02a |
| 0.1 | 0.00 ± 0.00d | 0.00 ± 0.00c | 0.87 ± 0.04a | 0.76 ± 0.06a | 0.87 ± 0.10a | 0.59 ± 0.08a | 0.85 ± 0.06a | 0.75 ± 0.02a |
| 1.0 | 0.00 ± 0.00d | 0.00 ± 0.00c | 0.49 ± 0.05b | 0.59 ± 0.05b | 0.83 ± 0.07a | 0.52 ± 0.13a | 0.88 ± 0.04a | 0.85 ± 0.04a |

**Notes.**

Data are shown as the mean ± standard deviation (SD) of four replicates. Different lowercase letters indicate significant differences between solutions of different concentrations ($P < 0.05$).

[a]ND: not detected.
allelopathic effects of extracts from different parts of *G. parviflora* on *M. sativa* and *A. sativa*. The Pearson correlation coefficients of SAE values of the whole plants and roots, stems and leaves on *M. sativa* were 0.850, 0.777 and 0.973, respectively, and that on *A. sativa* were 0.591, 0.483 and 0.761. The results indicated the allelopathy of leaves had stronger correlation with the whole plants of *G. parviflora*. It was also found that the allelopathic effect of the extracts from leaves of *G. parviflora* on the GR of *M. sativa* and *A. sativa* was significantly higher than that from roots (Fig. 1). In particular, the proportions of phytotoxic CHDA in the extracts of roots, stems, leaves and whole plants were 0.74%, 34.69%, 37.39% and 29.65%, respectively (Fig. 5). The lack of CHDA in roots of *G. parviflora* might be responsible for this difference. Unlike this study, *Wang et al. (2012)* found that the allelopathic effect of the extracts from roots of *G. parviflora* on the GR of *G. japonicum* was the highest. This might be because different recipient plants tested might have a different tolerance to the same allelochemicals or there are other allelochemicals at work.

This study quantitatively analyzed the chemical composition of the extracts from roots, stems, leaves and whole plants of *G. parviflora* for the first time (Fig. 5; Table 1). It was found that the highest content component in the extracts of *G. parviflora* was CHDA or trehalose, which were much higher than other components (Fig. 5) and might play an important role in its growth regulation. The herbicidal activity assay has also confirmed that CHDA began to inhibit the seed germination of *D. sanguinalis* at the concentration of $1.0 \times 10^{-4}$ g mL$^{-1}$, but did not affect the germination of *M. sativa* and *A. sativa* when the dosage was increased by 10,000 times (Table 2). It was precisely due to the sensitivity differences between crops and weeds that CHDA has the potential to be developed as a bioherbicide for artificial grassland. In fact, CHDA is an aliphatic dicarboxylic acid and an important intermediate in chemical and pharmaceutical industries (*Huang et al., 2006*). In addition, a complex crystal containing CHDA has been reported to have antibacterial, antioxidant, and cytotoxic properties (*Shanjitha et al., 2022*). Nevertheless, there have been no reports of its use in weed control. Our preliminary research indicated this substance is prone to causing fungal growth in plants at high concentrations, which may be the reason why it has not been utilized temporarily. Although the mechanism of CHDA is still unclear, it is worth further study for its the effective application. Coincidentally, CHDA has a similar chemical structure to commercialized plant growth delayer, prohexadione, which mainly interferes with the final step of gibberellin synthesis (*Yang et al., 2002*). Meanwhile, 4-hydroxy-5-alkoxy-1,2-cyclohexanedicarboxylic acid, similar to CHDA, was identified as a novel glycosidases (glycoside hydrolases) inhibitor (*Brazdova et al., 2009*). These evidences may be enlightening for the reason why CHDA inhibited the germination of *D. sanguinalis*. To be sure, the preparation technology of herbicides, the activity of more weed varieties, the risk assessment of the environment and many other issues should be considered in the application of CHDA in the future. On the contrary, trehalose was not toxic to the four tested plants, but it could resist various environmental stresses in a variety of plants and microorganisms (*Hasanuzzaman et al., 2020*; *Onwea et al., 2022*). For example, the content of trehalose is high in drought tolerant plants such as *Selaginella tamariscina* (P. Beauv.) Spring (*Goddijn & Dun, 1999*). Exogenous trehalose has been reported to promote the

synthesis of carbohydrates in plants such as *Zea mays* L. and help resist unfavorable growth environments such as high temperatures (*Zhang et al., 2022*). Therefore, the high content of trehalose in *G. parviflora* may be an important help for its invasion.

In recent years, many studies also tried to explore phytochemical composition in *G. parviflora*. At present, saponins and mucous compounds, flavonoids, aromatic esters, and vitamin C have been found in ethanol extracts from *G. parviflora* (*Ali, Zameer & Yaqoob, 2017*; *Mostafa et al., 2013*). In general, phenolic acids and polysaccharides were the main allelochemicals in the extracts from *G. parviflora* (Fig. 5). This study only explored the herbicidal activity of a phenolic acid and a polysaccharide with the highest content, and other components in the extracts may still have inhibitory effects. For example, 3,5-dicaffeoylquinic acid (isochlorogenic acid A) was considered to be the key allelochemical in *A. argyi* with broad herbicidal activity (*Chen et al., 2022*), ranking 11th in the content of the whole extracts of *G. parviflora*. Meanwhile, the content of 3,4-dicaffeoylquinic acid (isochlorogenic acid B) with similar structure was higher (Table 1), which deserves special attention in further research. However, the drawbacks of allelopathic substances released by plants cannot be ignored. The accumulation of allelochemicals might not only cause continuous cropping obstacles, but led to ecosystem disturbance (*Mitrovic et al., 2012*). Additional studies are needed to better apply allelochemicals to weed control.

## CONCLUSION

In summary, the extracts of *G. parviflora* exhibited allelopathic effects on the GR, GP, height and fresh weight of seedlings, chlorophyll content of *M. sativa* and *A. sativa*, and the allelopathic effect of extracts gradually strengthened with increasing concentration. Almost all extracts with a concentration of 0.075 g mL$^{-1}$ had significant inhibitory effects on the recipient plants. The different inhibitory effects of extracts from different parts of *G. parviflora* on recipient plants were as follows: the root was the weakest, and the whole plants was the strongest, with the values of SAE at 0.100 g mL$^{-1}$ on *M. sativa* were $-0.12$ and $-0.40$, respectively. In addition, the main component of extracts in *G. parviflora*, CHDA, effectively inhibited the germination of *D. sanguinalis*, but failed to affect *A. retroflexus*, which might have the potential to control gramineous weeds in cultivated forage fields. However, this study only explored the two components with the highest content, more studies should be carried out on the herbicidal activity of other components, such as 3,4-dicaffeoylquinic acid, with a view to achieving more comprehensive analysis of its release pathways and mechanisms of action on other plants.

## ACKNOWLEDGEMENTS

We are grateful to the staff of Yunnan Academe of Grassland and Animal Science in China for their enthusiastic assistance in seed collection.

### Funding

This work was supported by the National Natural Science Foundation of China (No. 32302407), the China Agricultural Research System (No. CARS-34) and the Doctoral Scientific Research Startup of Qingdao Agricultural University (No. 6631123002). The funders had no role in study design, data collection and analysis, decision to publish, or preparation of the manuscript.

### Grant Disclosures

The following grant information was disclosed by the authors:
The National Natural Science Foundation of China: No. 32302407.
The China Agricultural Research System: No. CARS-34.
The Doctoral Scientific Research Startup of Qingdao Agricultural University: No. 6631123002.

### Competing Interests

The authors declare there are no competing interests.

### Author Contributions

- Shipu Cheng performed the experiments, prepared figures and/or tables, and approved the final draft.
- Fanru Xu analyzed the data, prepared figures and/or tables, and approved the final draft.
- Zhiyong Lu performed the experiments, prepared figures and/or tables, and approved the final draft.
- Huairui Xu analyzed the data, authored or reviewed drafts of the article, and approved the final draft.
- Mengqi Cai analyzed the data, authored or reviewed drafts of the article, and approved the final draft.
- Juan Sun conceived and designed the experiments, authored or reviewed drafts of the article, and approved the final draft.
- Yufang Xu conceived and designed the experiments, prepared figures and/or tables, and approved the final draft.

### Data Availability

The raw measurements are available in the Supplementary Files.

### Supplemental Information

Supplemental information for this article can be found online at http://dx.doi.org/10.7717/peerj.19378#supplemental-information.

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
