# Peer review of "Allelopathic effects and composition of aqueous extracts from different parts of *Galinsoga parviflora* Cav. on *Medicago sativa* L. and *Avena sativa* L"

_PeerJ, doi:10.7717/peerj.19378_

## Round 0.1 · original submission · Major Revisions

Allelopathic impacts of the plant extracts obtained from the invasive plants on field crops or forage plants are significant in preventing yield and quality losses. Your study evaluates various aspects of allelopathic chemicals of G. parviflora, including practical ways. Although your work contains valuable information about allelopathic substances of G. parviflora, it does not fully meet some requirements of our journal. Therefore, it is essential to address certain technical details to enhance the article further. I strongly recommend carefully reviewing the reviewers' suggestions and thoughtfully considering each recommendation. If you disagree with any suggestion, it would be helpful to provide clear, well-reasoned justifications for your viewpoint.

Reviewer 1 ·

Basic reporting

The manuscript investigates the allelopathic effects of aqueous extracts from different parts of the invasive plant Galinsoga parviflora on two forage species, Medicago sativa and Avena sativa. Using germination rate, seedling growth, and chlorophyll content as indicators, the study finds that extracts have concentration-dependent inhibitory effects, with the strongest inhibition from whole-plant extracts. Additionally, metabolomic analysis identifies key allelopathic compounds, such as 1,4-cyclohexanedicarboxylic acid (CHDA), which exhibits selective herbicidal activity against weeds like Digitaria sanguinalis while sparing forages. The findings suggest potential applications for CHDA in natural weed management and provide insights into the allelopathic mechanisms of G. parviflora.
The manuscript in its current form is classified as a "major revision" because it requires several significant improvements in methodology, statistical analysis, and figures. However, the importance of the data does not warrant rejection.
The English language is at an intermediate level, but for a more fluent and professional readability, it would be worth doing some language proofreading.
The literature cited is sometimes outdated or not fully relevant to the research question.
The colors and labels of the figures are not always clear, especially in the case of interaction effects. The use of a color scale or better explanatory captions would be recommended and a larger font size would be needed (e.g. Fig. 5). The figure captions lack basic statistical data.
The research questions should be better explained and what the assumptions/hypotheses were should be defined.

Experimental design

The research questions should be better explained and what the assumptions/hypotheses were should be defined.
The materials and methods need to be improved, many details are missing that do not allow for reproducibility. The method of preparing the extracts is described in detail, but the rationale for choosing the concentrations is not clear. Why were these concentrations chosen?
The tools used are not well described or named. It is not mandatory, but the EC number of the chemicals used can also be indicated.
The statistical analysis of the interaction effects (concentration and plant part) was not given sufficient attention, although this would be important for the evaluation of the results.
Although the use of ANOVA tests is appropriate, the text does not explain the interpretation of the results in sufficient detail. For example, the significance of the "P × C" interaction is not contextualized.
How was the normality test performed? This is important because the appropriate statistical analysis must be selected based on it. At what p-value should the differences be considered significant?
For the sake of data transparency, it would be worthwhile to present detailed raw data in a separate appendix.

Validity of the findings

Introduction: Overall, the introductory topics require a more detailed explanation. The context of the introductory part presents the problem of invasive species well, but does not sufficiently address the specific novelty of the research. For example: Why is it important to study Avena sativa and Medicago sativa, Digitaria sanguinalis and Amaranthus retroflexus against invasive plants? Why were they chosen as test plants? Plese introduce them.
Results: It should focus on the most important trends. The interpretation of the results is sometimes too general and not always related to the main research questions. For example, further explanation of the mechanism of action of 1,4-cyclohexanedicarboxylic acid would be warranted. Also, p-values should be reported, not just whether something is significant or not.
Discussion: This part needs a stronger rewrite. Some statements are too general or not well supported. For examle: L 312-313. How does this relate to the current investigation? What are you trying to imply by this? Although some previous studies are cited in the discussion, it is not always explained how the results are similar or different from those presented in the literature. The mechanism of action of 1,4-cyclohexanedicarboxylic acid is mentioned in passing, but more detail would be warranted, especially to understand how it affects the biological processes of weeds and source plants. Although it is mentioned that the results may contribute to the development of weed control strategies, they do not address the practical challenges that may hinder their application (e.g., stability of extracts, environmental effects). The discussion is sometimes redundant, e.g. in the repetition of results.
Conclusions: The conclusions of the study do not fully reflect the data presented, e.g. the development of specific strategies against invasive plants.

Additional comments

L19: suggested correction: Galinsoga parviflora is a high-risk invasive plant that seriously threatens the..
L24: suggested correction: The germination rate (GR), germination potential (GE), seedling height, fresh weight, and chlorophyll content of..
L30: It is not really clear how these two species got here. (Digitaria and Amaranthus)
L32: Is the semi-sentence necessary? This could be more of an introduction to a discussion, as it seems to want to explain the results. It should be deleted or rephrased. (The results suggest that)
L46: The first mention of the scientific name of a species must include the name (or abbreviated name) of its describer. E.g.: Galinsoga parviflora Cav.; Avena sativa L. Please check these in the full text.
L47: The genus name is always italicized “Galinsoga”.
L71: ‘s’ is missing: Trifolium repens
L86: What was the planting medium and what basic parameters did it have? What were they watered with, and at what intervals?
L90: What was the plant material crushed with and to what size?
L102: Was the square culture dish plastic? The type of product, name of the manufacturing company is missing.
L155.156: ..phase A was.. ..phase B was..
L111: Why is germination potential abbreviated as GE? Wouldn't it be simpler to abbreviate it as GP? G. parviflora shouldn't be confusing because it's not used to fully abbreviate the species.
L148-149: how it was centrifuged (centrifuge type, manufacturer)?
L294: The ful stop is in bold. “M. sativa”
L318: The yer is missing: Wang et al.

·

Basic reporting

The article meets the requirements of the journal regarding the necessary standard sections and their presentation. Figures and tables correspond to the content of the article and the additional file data presented, which were properly attached to the article itself. The authors have formed the research objectives, but there is no clear research hypothesis. The article contains properly formatted references to other sources of research that are relevant to the main focus of the article. In general, the submitted publication should be assessed as a self-sufficient publication component, which, however, requires some correction and improvement. In particular, to actualize the purpose and objectives of the research, the authors of the article used the statement of the invasiveness of this weed species from the position of a quarantine weed with high adaptive potential. This is a specific type of weed that has a fairly wide range of biological and allopathic adaptations to form a dominant vitalistic tactic in the lower and middle tier of the agrocenosis of the respective crop. However, in my opinion, it is necessary to detail the levels of crop losses and the corresponding economic or environmental losses by adding materials from such studies as:

Riemens, M.M. and R.Y. van der Weide. 2008. Biology and Control of Galinsoga parviflora, overview of literature survey. Wageningen: Plant Research International. Note 576, 32 pp.
Damalas C.A. 2008. Distribution, biology, and agricultural importance of Galinsoga parviflora (Asteraceae). Weed Biol. Manag. 8: 147–153.
Mostafa I, Abd El-Aziz E, Hafez S, El-Shazly A. Chemical constituents and biological activities of Galinsoga parviflora cav. (Asteraceae) from Egypt. Z Naturforsch C J Biosci. 2013 Jul-Aug;68(7-8):285-92. PMID: 24066513.
Paula, D. F. d., Silva, E. M. G. d., Silva, L. B. X. d., Lima, A. d. C., Billu, P. B., Reis, M. R. d., & Mendes, K. F. 2022. Sustainable Control of Galinsoga parviflora with Oxyfluorfen, Flumioxazin, and Linuron Application in Two Soils Cultivated with Garlic. Sustainability, 14(24), 16637. https://doi.org/10.3390/su142416637

The digital detailing of the herbological potential of weeds will be appropriate both in the global dimension and in the application to the provinces of China. This will increase the scientific interest and future citation of the article.
The statement in line 58 “released by a certain plant” should also be modernized by adding “by root exudates, transpiratory secretions and the results of the decomposition of their residues in the soil” (which would be fully consistent with the classics of allelopathic research (Willis, Grodzinsky)). Tape 67 on radish can also be supplemented by Tsytsiura Y., Sampietro D. Allelopathic Effects of Annual Weeds on Germination and Seedling Growth of Oilseed Radish (Raphanus sativus L. var. oleiformis Pers.). Acta Fytotechnica et Zootechnica. 2024. Vol. 27. no. 1. P. 77-97.

Trifolium repen(s) is omitted in tape 71.
Lines 75 and 76, in my opinion, should be supplemented with other regions of the world, since alfalfa and oats are also cultivated in other parts of the world (and are among the main crops by share in crops) with a temperate climate and in conditions of unstable moisture where this weed species is a real threat to feed quality.
After tape 81, it should be noted what these studies will allow to achieve in the agrotechnological direction from the point of view of weed control in agrocenoses.

Results. The description of the data should still be more specific, in particular, the concepts of more or less are concepts of a more general nature of the analysis, and a ratio could be used to detail the effects. The same applies to the SAE score. This will allow us to move away from the general effects of the description to the specifics and real indication of allopathic effects.

Discussion. It would be desirable to add the authors' assessment of the level of complementary allelopathic action of the total extract from the whole plant in comparison with individual parts, again using the approaches of coefficient analysis. In particular, there is no clear explanation for the effects of an increase or decrease in test parameters in comparison of individual plant parts before its use as whole plant extracts. In addition, the analysis of the isolated allelopathic substances in the discussion is presented only in separate groups and does not have specifics on the main dominant chemical compounds for all parts of the plant. I will not be specific, but there are a number of publications on the allelopathic potential of the compounds presented in Figure 5. Because the whole picture of the potential of G. parviflora in terms of the formation of adaptive allelopathic effects in agrocenosis is not entirely clear. It would be desirable to add the results of the study of this weed species on related test objects to ensure the search for analogies and resulting explanations. This creates a somewhat mosaic impression of the results, but they are presented in the tables and are quite interesting and relevant. Also, in my opinion, it would be desirable for the authors to add, if possible, a brief description of the changes in the structure of allopathically active substances in different parts of plants and the specifics of the impact of this fact on the adaptive component of the allelopathic potential of G. parviflora.

References. In general, it covers the general aspects of the introduction and discussion of the results, but could be improved by adding the references I provided in the previous sections of the review.

Images and Tables. Figures have not been inappropriately manipulated. However, the captions in small print are difficult to read in the general pdf version of the article. If possible, increase the font size to the maximum allowable level according to the design requirements, which will significantly improve the perception of the article. In addition, the authors may consider converting Figures 1-4 to color as well.

Raw data have not been inappropriately manipulated

Comment on language and grammar issues. I am not a native speaker, but the basic terminology and indicator indicators for allopathic analysis are presented correctly, as well as the biochemical and biospecies components. The text sometimes contains too short and simple sentences that could be combined with a general abbreviation of both, but this is my personal opinion.

Experimental design

Original primary research within Aims and Scope of the journal.
Methods should be described with sufficient information to be reproducible by another investigatorю In particular, the authors used a modified method of forming allelopathic extracts through laboratory-controlled germination of collected seeds of a segetal species that had previously grown to the seed stage in the agrocenosis of the base recipient. And according to the process scheme, everything is correct. However, I have some methodological questions as to why the option of direct field selection of the relevant parts of the plant during the flowering phase, which for most species is the period of maximum allelopathic activity and is recommended for the selection of plant material, was not used (Fujii, Y., & Hiradate S. (2007). Allelopathy: New Concepts And Methodology. CRC Press.; Zhang Z, Liu Y, Yuan L, Weber E, van Kleunen M. Effect of allelopathy on plant performance: a meta-analysis. Ecol Lett. 2021 Feb;24(2):348-362. doi: 10.1111/ele.13627. Epub 2020 Oct 21. PMID: 33085152., Shan, Z., Zhou, S., Shah, A., Arafat, Y., Arif Hussain Rizvi, S., & Shao, H. (2023). Plant Allelopathy in Response to Biotic and Abiotic Factors. Agronomy, 13(9), 2358. https://doi.org/10.3390/agronomy13092358 and others).

Therefore, it is advisable to indicate in line 89 which phenological phase or life stage of the plants was reached on day 45. Given the acceptability of this method, I wonder why the weed seeds were sampled from alfalfa crops and not from both test crops and then proportionally mixed to form a test sample. The use of controlled germination conditions according to the basic research (Weidenhamer, J.D. (2008). Allelopathic Mechanisms and Experimental Methodology. In: Zeng, R.S., Mallik, A.U., Luo, S.M. (eds) Allelopathy in Sustainable Agriculture and Forestry. Springer, New York, NY. https://doi.org/10.1007/978-0-387-77337-7_6) forms certain differences in the plant's allelopathic portfolio, which requires appropriate adjustments for the subsequent practical use of the results. In addition, this species is characterized by certain features of the period of biological dormancy of seeds. Did the authors take this fact into account?
Tape 110 was used ISTA, 1985, although ISTA (2020) could be used with the procedure. International rules for seed testing. Chapter 5: The germination test. International Seed Testing Association (pp. 5-58).
In line 119, please provide an explanation or reference for the 21 days (methodology).
In line 120, brief information on determining the wet and dry weight of the selected plant samples is desirable.
Tapes 176 and 177 are questions about taking into account the periods of biological dormancy of test weed seeds and the impact of this indicator on overall seed germination (Qasem, R.J. (2020). Weed Seed Dormancy: The Ecophysiology and Survival Strategies. IntechOpen. doi: 10.5772/intechopen.88015).
The grading system of SAE could be based on the grading system (0-0.25 Non-allelopathic (NA); 0.26-0.5 Moderately allelopathic (MA); 0.51-0.75 Highly allelopathic (HA); 0.76-1.0 Extremely allelopathic (EA)) according to Smith, O.P. (2013). Allelopathic Potential of the Invasive Alien Himalayan Balsam (Impatiens glandulifera Royle). A thesis submitted to Plymouth University in partial fulfillment for the degree of Doctor of Philosophy. 388 pp).
It is also necessary to indicate why these two components were evaluated for herbicidal activity from the general list of components identified during the analysis (a brief scientific justification is needed, since there are a number of interesting components from the list presented that deserve special attention in further research).

Validity of the findings

All underlying data have been provided; they are robust, statistically sound, & controlled. The conclusions are based on the main results of the research, but I would recommend that they be somewhat more specific: In line 357, indicate the interval of concentration change or the dynamic gradient of its change; in lines 358-359, give the gradient assessment I mentioned in terms of a ranked series of allopathic intensity of different parts of the plant; I would rephrase the conclusion in lines 360-361 in relation to certain weed species groups, since in this context the statement does not correspond to the test sample of the assessment objects. It should also be specified which components were evaluated from the point of view of herbological control.

Additional comments

The article makes a positive impression and contains novelty and is based on modern methodological research approaches. I recommend it for publication after the authors have made the corrections and main areas of improvement noted by me in accordance with the above sections of the review. I believe that research in the field of alelopathy in general and the authors' research presentation in particular is a general perspective of innovative agricultural technologies from the standpoint of controlling and improving the quality of products and the environment. Therefore, I wish the authors fruitful work and high knowledge-intensive results in this area.

---

## Round 0.2 · Minor Revisions

I appreciate your positive and constructive attitude toward the suggestions of reviewers. However, your article needs some revisions to improve before publishing. I suggest a thorough review of the reviewers' suggestions and a judicious consideration of each recommendation. If you find yourself in disagreement with any particular suggestion, it would be beneficial to provide clear and well-reasoned justifications for your perspective.

Reviewer 1 ·

Basic reporting

Dear Authors,
The authors have made significant revisions to the manuscript based on the reviewers' suggestions. However, some minor clarifications and further explanations are necessary:

The analysis of the CHDA mechanism of action remains incomplete. Although the authors refer to structurally similar compounds, it is still not entirely clear how CHDA affects weed species.
The ecological consequences of allelopathic compounds have not been fully addressed (e.g., degradation, environmental stability).
The rejection of the SAE (synthetical allelopathic effects) classification system has been well justified, but introducing a custom classification system could be beneficial.

Experimental design

No comment.

Validity of the findings

No comment.

·

Basic reporting

The article meets the requirements of the journal regarding the necessary standard sections and their presentation. Figures and tables correspond to the content of the article and the additional file data presented, which were properly attached to the article itself. In accordance with the previous review, the authors have made appropriate amendments to the introduction.
In particular, the harmfulness of G. parviflora is detailed, the specifics of its invasiveness from the point of view of dominance in the agrocenosis are summarized. The formation of the research hypothesis from the standpoint of the general theoretical framework of the allopathic profile of plants was also clarified. It is also positive that the authors take into account the relevance of this weed species in general for world practice and not only from the point of view of localization of the research site.
I consider the explanations of the authors and the corrections they made in the letter of response regarding the results section and the discussion of the results to be quite sufficient. There is still a certain specification for updating the structure of allelochemicals that play a dominant role in the analysis, but I believe that this would be subjective on my part. For these reasons, these clarifications are quite sufficient and detail the places that were somewhat controversial for me in the first version of the article.

Experimental design

Original primary research within Aims and Scope of the journal. Methods should be described with sufficient information to be reproducible by another investigatorю In particular, the authors used a modified method of forming allelopathic extracts through laboratory-controlled germination of collected seeds of a segetal species that had previously grown to the seed stage in the agrocenosis of the base recipient. And according to the process scheme, everything is correct.
All the comments made in my previous review have either been taken into account or given objective explanations with which I agree.
There is still some debate about the indexing of allopathic inhibition, but the authors are quite right to insist on the chosen inhibition option, and it is difficult to disagree.

Validity of the findings

I confirm once again that аll underlying data have been provided; they are robust, statistically sound, & controlled. The conclusions are based on the main results of the research. The clarifications provided and the corrections made to the amended version of the article satisfy me as a reviewer.

Additional comments

I am grateful to the authors for their careful work on the article and thank them for their explanations, as well as for their own position as scholars in defending their own opinions. As before, I wish them fruitful scientific activity and possible cooperation in the study of the allelopathic component of technologies.

---

## Round 0.3 · accepted · Accept

I would like to thank you for accepting the referees' suggestions and improving your article based on their suggestions. Your article is ready to publish. We look forward to your next article.

For instance, the Section Editor noted:

> Various English corrections should be made. Here are some of them that I noticed: L 17 “effects of” should be “effects on” L 33 “were” should be “being” L 36 “good” should be “strong” L 89 “to two” should be “on two” L 92 “source” should be “sources” L 108 “for to obtain” should be “to obtain” L 404 “as follows, the root” should be “as follows: the root” L 411 “action that mediate on” should be “action on”

Reviewer 1 ·

Basic reporting

I consider the corrections made by the authors to be appropriate, and I recommend the manuscript for publication.

Experimental design

No comment.

Validity of the findings

No comment.

Additional comments

No comment.